# Peri-Urban Community Attitudes towards Codling Moth Trapping and Suppression Using the Sterile Insect Technique in New Zealand

**DOI:** 10.3390/insects10100335

**Published:** 2019-10-09

**Authors:** Georgia Paterson, George L. W. Perry, James T. S. Walker, David Maxwell Suckling

**Affiliations:** 1School of Environment, University of Auckland, Auckland 1010, New Zealand; gpat568@aucklanduni.ac.nz (G.P.); george.perry@auckland.ac.nz (G.L.W.P.); 2The New Zealand Institute for Plant and Food Research Limited, Havelock North 4157, New Zealand; Jim.Walker@plantandfood.co.nz; 3The New Zealand Institute for Plant and Food Research Limited, Private Bag 4704, Christchurch 8140, New Zealand; 4School of Biological Sciences, University of Auckland, Auckland 1072, New Zealand

**Keywords:** eradication, public, biosecurity, *Cydia pomonella*, unmanned aerial vehicle

## Abstract

New, more socially-acceptable technologies are being developed to suppress horticultural pests, because suppression is technically difficult with current technologies, especially in urban areas. One technique involves the release of sterile insects to prevent offspring in the next generation. This technology involves aerial or ground release systems, but this could also create issues for the public. This study investigated community perceptions of a recently-introduced response to codling moth control in New Zealand—Sterile Insect Technique (SIT). Community attitudes to SIT were examined in Hastings, New Zealand, in April, 2018. Eighty-six detailed interviews were undertaken with a random sample of households. This community was very willing (98% agreement) to host a sex pheromone trap in their gardens, and condoned regular visits to monitor traps. Attitudes to SIT were very positive (98% in favor). Once explained, the concept of using unmanned aerial vehicles to deliver sterile insects was also acceptable (98%) to the community. Use of unmanned aerial vehicles to release sterile insects during a hypothetical incursion response of an exotic fruit fly was also supported at 98% by respondent householders. Investigation of community attitudes can be valuable to guide practitioners in determining suitable technologies before an area-wide programme is launched.

## 1. Introduction

Globalisation is leading to an upsurge in insect invasions into new jurisdictions [1,2], leading to attempts to prevent their establishment [3]. Invasive species are an increasing threat to food production and ecosystem health as a consequence of global trade and climate change [4,5]. Failure to control invasive arthropod species can jeopardise the profitability and sustainability of production [6,7] and can have devastating ecological effects [8,9]. However, the apparently unstoppable increase in globalisation of trade together with climate warming results in a significantly increased risk of pest species incursion and establishment [10]. 

While nearly 100 countries have undertaken one or more insect pest eradication programmes in response to specific threats, some jurisdictions have been involved in many responses [11]. New Zealand’s geographic isolation, in combination with a relatively stringent plant quarantine and biosecurity system with active post-border surveillance for exotic pests has meant the relative absence of many potentially devastating fruit pests. Ironically, the greater the move towards residue-free production and Integrated Pest Management [12], the greater the exposure to incursion risk of devastating exotic pests that are difficult to control [13]. Further, social researchers working on invasive fruit flies have pointed out that “Social mechanisms underpinning collaborative approaches to pest management are as important as the biological control of the pest” [14].

An increase in incursions has led to research on novel technologies to respond to dispersed insects whose exact location is uncertain within an area. This has led to adoption of the concept of Area-Wide control. Correspondingly, there is an increasing need for understanding of community perceptions in relation to these new technologies [15]. Combinations of benign tactics can offer quicker and more cost-effective suppression of pest populations [16,17]. 

In particular, autocidal methods are preferred where synergy is sometimes possible between methods [16]. In particular, the Sterile Insect Technique (SIT) can help to suppress populations in proportion to the ratio of mass released and residual wild populations, through the mass rearing and release of sterile insects [18,19,20]. A multiple tactic approach including SIT [20] is being evaluated for the local eradication of codling moth in New Zealand (*Cydia pomonella*) [21], an insect which has been present for 100 years [22] and the target of IPM by mating disruption over large areas for many years [12]. Export-focused and documented suppression of codling moth has involved submission of spray diaries and insect counts, and a shift towards more benign control tactics [12]. Orchard populations of this pest have been dramatically reduced by mating disruption over the last decade [12]. A Canadian initiative to develop the SIT for codling moth [23] led to the development of a factory and long-standing release of moths into orchards for population suppression [24]. In New Zealand, we have targeted an isolated sub-region (Central Hawke’s Bay) to test the effects of the SIT on private orchard land. This combination of the SIT using insects imported weekly from British Columbia, together with sex pheromone dispensers for mating disruption covering a series of large (50–100 ha) orchards has greatly reduced populations of codling moth since 2014 [21]. We set out to explore the peri-urban sources of the insect.

Experience of community responses to insect eradications in New Zealand [25] and California [26] led us to investigate community attitudes in the context of codling moth suppression in a peri-urban area of a major export apple production region. A successful eradication programme requires a holistic and multidisciplinary approach [27], which includes involvement of communities living in close proximity to the pest and potentially within the area targeted for control [28]. In our case, these are peri-urban communities located not far from horticultural enterprises, which we seek to better understand through dialogue. As pest control is a key requirement in New Zealand’s efforts to conserve its biodiversity and sustain primary production, a public that is informed on the policies and practices involved can increase opportunities to influence decision making and the selection of methods used. This requires those designing environmental policy to recognise public concerns about pest control methods and practices. 

The management of invasive species can lead to social conflicts amongst affected stakeholders [29]. The complexities involved in many programmes lead to uncertainty among communities and other stakeholders. Some of these uncertainties are the result of a lack of access to timely information [30]. While conflicts and problems are not always avoidable, with meticulous planning and community collaboration, other researchers report that this approach can reduce the likelihood of these issues arising [31]. As many host apple or walnut trees are present in private gardens in the Central Hawke’s Bay region, communication with the public is essential to inform them of events, seek their co-operation in accessing traps and explain the biological processes involved in attempting to control codling moth using this technique. The community of Hastings has had a long history of exposure to horticulture and knowledge of pest and disease management practices.

We undertook a study to gauge community responses to proposed pest suppression or eradication technologies for the codling moth. The purpose of applying these technologies is to suppress, or possibly eradicate, codling moth from peri-urban areas in Hawke’s Bay. If successful, controls of codling moth in apple orchards targeting export markets would not be needed, nor would fumigation for market access (desirable, as the current ozone-depleting fumigant methyl bromide is to be withdrawn in 2021, under the 1987 Montreal Protocol on Substances that Deplete the Ozone Layer) [32].

The orchard-based SIT pilot programme [23] has led to significant reductions in pest numbers in the Hawke’s Bay. A large exporting company is now funding the application of SIT to a further area covering approximately 500 ha of orchard in the Hawke’s Bay in 2019 [33]. However, if the concept were to be expanded to the main growing region (including public areas) of the Heretaunga Plains, it would require wider community acceptance. As a result of its technical success, the team are considering the implications of proposing a truly area-wide management programme. The results of the survey described in this paper will inform that assessment.

## 2. Methods

We assessed public engagement in codling moth eradication by questionnaire surveys of occupants of dwellings in the Hastings district in the North Island of New Zealand (Figure 1). The survey was designed, in part, to answer questions raised by scientists from The New Zealand Institute for Plant and Food Research Limited currently testing the SIT on private orchard land, using novel unmanned aerial vehicles for insect release. The attitudes, values and beliefs of the Hastings community were sought on a range of topics including their concern about pest species, their relationship to horticulture, whether they had a host tree on their property and their attitudes to proposed control methods involving traps and the SIT.

### 2.1. Ethic

Surveyors visited householders in a random sample of the Hawke’s Bay town of Hastings (Figure 1), under approval from the University of Auckland Human Participants Ethics Committee on 05/04/2018 (Protocol number 020222). The Participant Information Sheet (PIS, Appendix A) provided background on the research and the purpose of the study. Undertaking the survey implied participant consent. The participants were informed about the intended benefits of taking part in this research, including increased engagement of the community in eradication processes, their effects and outcomes; opportunities to voice opinions and concerns about pest control activities; and access to information about control processes to inform understanding.

### 2.2. Selection of Sample Size

The 2013 census estimated a total of 27,042 occupied dwellings in Hastings, 11,322 of which are within the selected study area [34]. In this area, residents at 86 dwellings were surveyed. Sample size is one of a few inter-related characteristics in the design of a study that has the ability to identify the significance of differences in relationships and interactions. Bartlett et al. [35] suggested that for a population size of around 10,000 and margin of error at 0.3, the lowest sample size selected should be 83, which is very close to our sample size. However, an acceptable margin of error for social research is usually 5% for categorical data and 3% for continuous data [36]. Thus, the limited sample size and infeasibility of increasing it within the resources and timeframe available suggest that caution should be taken in interpreting results statistically. However, results may provide useful indications of community perceptions.

### 2.3. Selected Study Site

The study was carried out in the region where pheromone traps had been set up to survey codling moth populations (Horner, Paterson et al. unpublished data). The dwellings included in the community study were selected to be c. 300 m away from each other and from pheromone traps.

### 2.4. Questionnaire

A questionnaire was developed to evaluate social acceptance of the proposed application of the SIT (Appendix A). The survey was carried out in person and data were collected during the visit to ensure responses were obtained. The interview began with a short introduction to the research and the research institutions involved. An overview was provided of privacy and ethical obligations established by the University of Auckland’s Human Participants Ethics Committee; anonymity was assured but it was explained that data cannot be withdrawn once the questionnaire has been submitted (Appendix A). It was then explained to the participants that Hawke’s Bay growers already use sex pheromones to control moths in apple orchards to ensure export fruit are free of chemical residues. The fact that these harmless practices support the economy and avoid the use of insecticides was also highlighted. Participants were made aware of the presence of codling moths in the peri-urban area and were told that this team was proposing to release sterile codling moths to control the remaining wild populations. Because of an interest in the spread of information in the horticulture sector about biosecurity, questions included whether the respondent had family that worked in horticulture. We pursued this with the respondent’s level of concern about mammalian pests and invasive insect species, whether they were happy to have monitoring traps set up, whether there were any apple or walnut trees on the property, and their acceptance of the use of unmanned aerial vehicles for insect release.

### 2.5. Data Analysis

The ordinal survey response data (1–5 scale: 1 = not very concerned, 5 = very concerned) were suitable for non-parametric statistical analysis. Kruskal–Wallis tests were performed on the data to test for main effects from gender and presence of a family member in horticulture, and on levels of concern over invasive species. All statistical analyses were conducted using Minitab 18.

## 3. Results

### 3.1. Age and Gender

According to the most recent census (2013), 52% of Hastings residents are female and 48% male [34]. Of the 86 respondents, 63% (*n* = 54) were female and 37% (*n* = 32) were male. The greater response from females was a reflection of presence in the households surveyed and willingness to participate. One male and ten females expressed initial reservations (Q10) but all were satisfied with the final response to open questions.

### 3.2. Concern about Invasive Species

Householders surveyed were concerned about invasive pests in general. A greater proportion of female participants than males expressed concern about invasive insect species (Figure 2), and this was significant by Kruskal–Wallis test (*H* = 17.33, *p* < 0.001).

There was a significant difference in the scores by Kruskal–Wallis test (*H* = 7.58, *p* < 0.01) for mammalian versus insect pests, with concern for mammalian pests being slightly higher than for insect pests; 51% of respondents categorized their concern for invasive mammalian pests as 5 (the highest possible score), whereas only 34% of respondents categorized their concern for invasive insect pests as 5. None of the public categorized their concern of invasive mammalian pests as 1 (the lowest possible score) whereas 5.8% assigned insect pests to this category (Figure 3).

### 3.3. Family in Horticulture

Employment in the Hastings district is centered on the horticultural and agricultural sectors. An estimated 21% (*n* = 8310) of the employed Hastings population worked in the agriculture, forestry and fishing industry at the time of the most recent estimate [34]. About a quarter of participants had a family member employed in horticulture (26%), reflecting the wider New Zealand population. The high level of concern in response to invasive insect species was significantly influenced by the presence of a family member in horticulture (*H* = 4.87, *p* < 0.05). Respondents with a family member in horticulture had a greater concern for invasive insect pest species; 58% giving a 5 rating and the remainder a 3 or 4 rating (Figure 4).

### 3.4. Pheromone Traps and Host Trees

When asked whether they would allow a pheromone trap to be set up on their property in the future, 98% of the survey participants answered yes. Of the 86 participants, 19% had a host tree on their property, and of those 81% had an apple tree and 19% had a walnut tree. Participants with properties that had a host apple or walnut tree, infested with codling moth or not, were asked whether they would consider removing their tree in exchange for an incentivized gift. Three-quarters of respondents (12/16) agreed to have the host tree removed in exchange for another non-host fruit tree or seedling. However, some (*n* = 4) were less agreeable to having their tree exchanged for another tree.

### 3.5. Response to Unmanned Aerial Vehicles

It was explained to participants that the team was proposing that sterile moths to control pest codling moths in the Hastings locality be released using a small, lightweight unmanned aerial vehicle (UAV) with a wing span of 2.5 m. This technique is already in use over orchards in the region [23]. The participants were then asked their views on the use of UAVs, to rate how comfortable they were with the idea, and whether they had any reservations about this method. Many participants were concerned about their privacy and asked whether the UAVs had cameras attached to them (Appendix A). Once it was explained how the UAVs would work, 98% of participants were supportive of the use of UAVs for insect release.

Respondents were then asked if they accepted this technology being used during an emergency fruit-fly or similar insect pest response. Most (98%) were supportive of an emergency release being carried out, and only 2% declined support.

Participants were then told that UAVs can be based on helicopters with eight rotors (“octocopters”) or have fixed wings (like a small plane). The survey asked if participants were open to the use of octocopters, with most very supportive of their use (Figure 5). No member of the surveyed community was completely against their use; most responded positively, with 63% responding with a 5 to the use of UAVs and 51% responding with a 5 to the use of octocopters.

### 3.6. General Community Response

Open-ended comments were recorded at the end of the survey, including from ten women and one man expressing early reservations (Q10) and noted again at the end of the questionnaire. From the responses, it is clear that most people surveyed were very supportive of the proposed eradication methods once they understood the technical details sufficiently to judge them benign. Many people voiced concerns over health issues such asthma and eczema as a result of insecticide spraying in close proximity to their homes and were, therefore, very supportive of sterile insect releases (Table 1). Some members of the community preferred the sterile insect method for ethical reasons as no organisms are directly killed. Questions and concerns were raised about whether, or how, the proposed actions would affect respondents’ pets or children. It was explained that the proposed methods should not affect them at all. This explanation apparently eased reservations about the proposed methods.

## 4. Discussion

### 4.1. Community Attitudes

The surveyed community was very concerned about invasive pest species, both mammals and insects. The slightly greater concern with invasive mammals may be because of wider media coverage of programs such as Predator Free New Zealand [37] and controversy over aerial use of mammalian poisons, such as sodium mono-fluoroacetate [38]. The community as a whole was largely supportive of emergency insect pest responses being carried out when necessary.

Attitudes and behaviors toward pest species are largely based on past experiences with the pest species [39]. For this reason, it was considered likely that those who had worked in orchards or had backyard apple trees would be more responsive to the proposed codling moth eradication methods. It was evident from the community response that many members of the public were aware of codling moth and its risk to the local economy and environment. There appeared to be more concern about invasive pest species among those with a family member in horticulture, compared with those with no family member in horticulture. This difference might be attributed to the prior knowledge that these respondents have regarding the risks that invasive pest species pose to the New Zealand economy and being aware of the proposed methods to control these pests.

### 4.2. Community Perceptions of Spraying

Previous studies suggest that historical use of insecticide spraying to control and/or eradicate invasive species is the method least preferred by the public. For example, focus groups comparing aerial *Bacillus thuringiensis*, aerial sex pheromone or aerial release of sterile insects, favored the SIT [28]. In the current study, it was evident from open-ended questions that a large proportion of respondents was firmly opposed to the use of insecticide sprays to control pest species. This opposition apparently arose not only from a lack of communication between orchard managers and the public about when these spraying events were to occur, but also because of the perceived health implications of the spray applications. A long history of spraying in the Hawke’s Bay orchards for codling moth and other pests has led to reported health issues for members of the community. The primary health problems described by members of the public during the community response to open questions concerned breathing and skin issues such as asthma and eczema.

If a community is not pre-warned and/or if necessary information is not adequately communicated before control activities and effects are seen or perceived, public mistrust and opposition can spread and intensify. For example, as stated in the New Zealand Ombudsman’s report on the complaints that arose from the aerial spraying of *Bacillus thuringiensis* Kurstaki (Foray 48B) in urban Auckland to control the painted apple moth (*Teia anartoides*) incursion, [40] (p.10) “The proportion of householders contacting the health service displaying irritability, frustration, anger and anxiety, outweighed those who suffered pre-existing mental illness…”. Therefore, although the negative outcomes for medical health may be low as in that case, lack of communication between the public and community caused community concern and conflict, which could have otherwise been avoided. Such concern effectively limits social capital in the form of license to operate. In Australia, trust in individuals advocating area-wide management and sterile insect technology of fruit flies was also seen as an important factor affecting their adoption, although trust between neighbors was also important [14]. 

Although there was a slightly more positive response to the use of fixed-wing UAVs than octocopters, in both cases the community was open to the proposed use of technology. It was hypothesized that the public would be less open to the use of octocopters for sterile releases as the research team considered that they may look more menacing then standard UAVs, but most respondents were very supportive of using UAVs to make sterile insect releases.

The drift of spray from horticultural land is stated to cause human health problems such as asthma, eczema and other skin rashes and watery eyes [41]. The responsibility for informing nearby residents of the date and time of spraying activity lies with the spray operator. A historical lack of communication over orchard spray events could have led to an aversion by householders to the use of insecticide sprays in the comments at the end of the survey (Table 1).

## 5. Conclusions

The participants in this survey were generally very positive about the proposed use of technologies and control methods, including the use of UAVs to release sterile insects. When members of the public are aware and well-informed about issues in the environment, they are more able to become more involved [8]. Invertebrate pests pose a major threat to New Zealand’s unique biota and economy, and there is increasing demand from the primary sector to protect crop production using acceptable, efficient and effective technologies. Increased awareness of the policies and practices involved in pest eradication and long-term management positions the public to engage in constructive dialogue about the effects of these technologies. The results from this survey are one contribution to informing environmental policy makers, regulators and pest management practitioners of the importance of community engagement and opportunities to improve engagement processes. The apple industry received the messages implicitly from the community questions about spraying and has indicated that it can use the information in outreach to the community.

## Figures and Tables

**Figure 1 insects-10-00335-f001:**
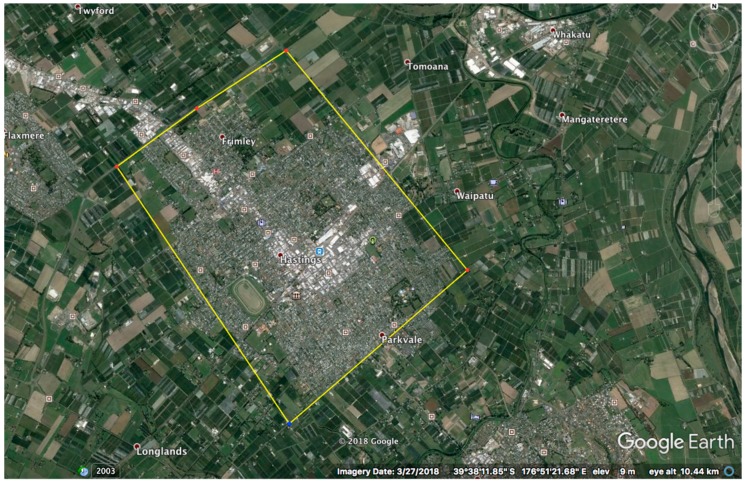
The location chosen for study of community attitudes to codling moth eradication and the Sterile Insect Technique using unmanned aerial vehicles was the indicated sample area of Hastings within Hawke’s Bay in New Zealand (peri-urban Hastings, 27,000 dwellings). Image courtesy Google Earth, 6/6/2019.

**Figure 2 insects-10-00335-f002:**
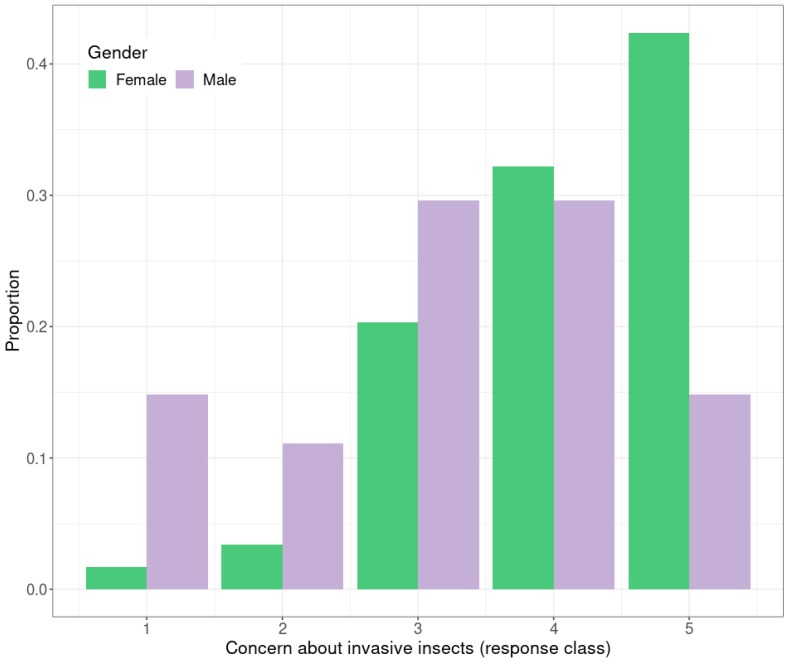
Proportion of respondents with different levels of concern about invasive pest insect species by gender. The scale ranges from 1 to 5 (1 = not very concerned, 5 = very concerned).

**Figure 3 insects-10-00335-f003:**
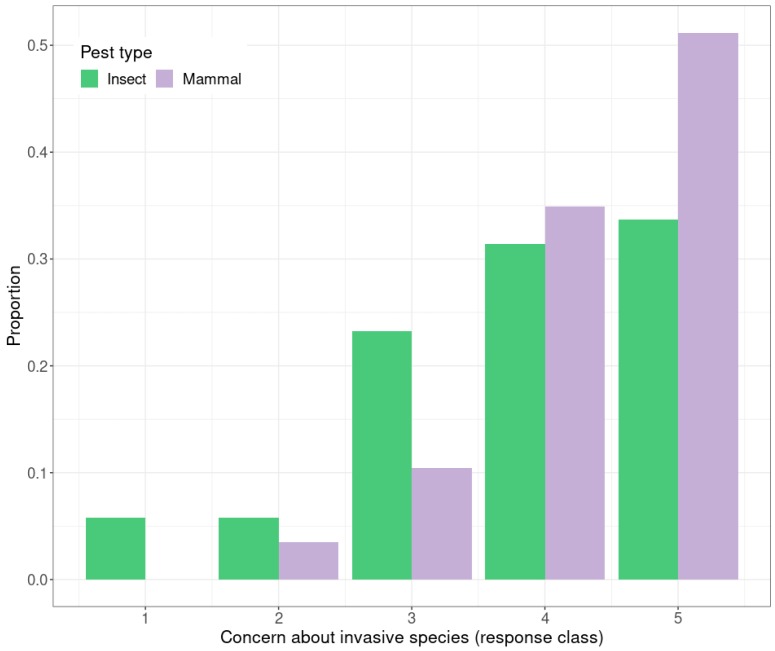
The proportion of respondents with different levels of concern about invasive mammalian pests compared with concern of invasive insect pests. The scale ranges from 1 to 5 (1 being not very concerned, 5 being very concerned) (*n* = 86).

**Figure 4 insects-10-00335-f004:**
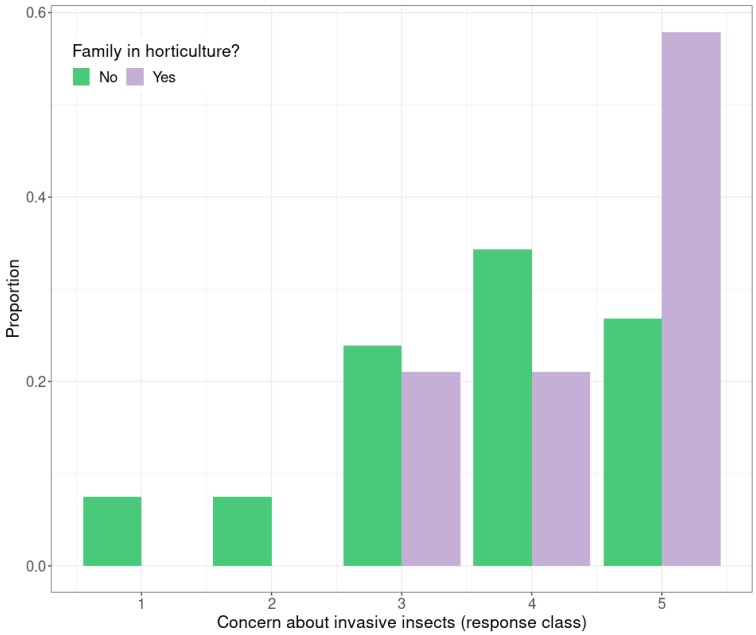
Proportion of survey respondents with family in horticulture compared with members of the public with no family in horticulture that have concern of invasive species. The scale ranges from 1 to 5 (1 = not very concerned, 5 = very concerned about invasive pest insects).

**Figure 5 insects-10-00335-f005:**
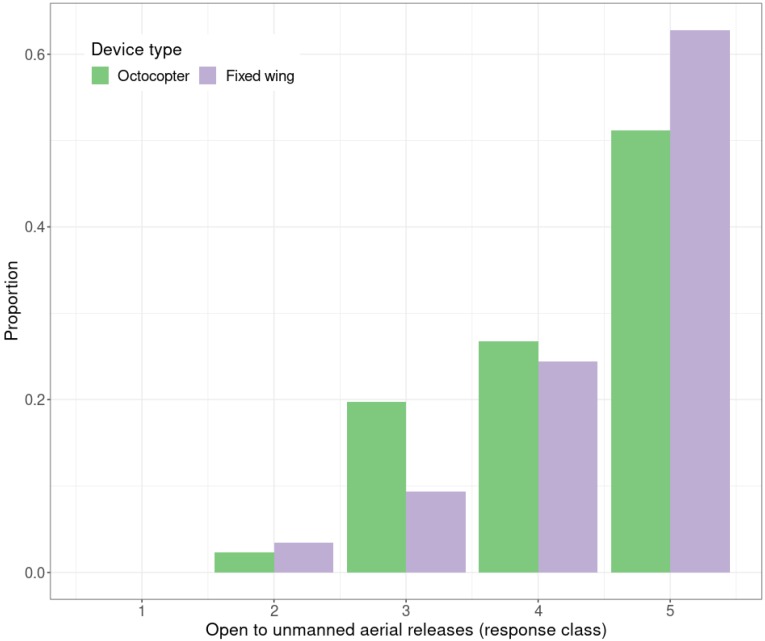
Response of the participants to the use of unmanned aerial vehicles that are of fixed-wing type, compared with the use of octocopters (helicopter type). The scale ranges from 1 to 5 (1 = not open at all, 5 = very open).

**Table 1 insects-10-00335-t001:** Questions raised by the community as a response to the proposed unmanned aerial vehicle methods, at the end of interviews, after: “*Do you have any further questions or comments regarding any of the proposed methods previously discussed?*”

Questions Raised
“Will there be cameras attached to these things? Will they be able to see me?” “Do they have cameras on them?” “Will this affect any other insects like butterflies?” “How many moths will get released and how often?” “Will this affect bee populations at all?” “Will this be bad for my children?” “Are there cameras attached?” “I don’t think this is important. What’s on my property?” “They aren’t toxic right? They can’t affect my toddler or cat?” “How do they know where to release?” “Do they have cameras?” “Will this affect my children? Is it toxic at all?” “How will this affect my cat? Will they bother him?” “Are there cameras on them? Will they record?” “Will this affect what’s in my garden?”

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
