# Peer review of "Peri-Urban Community Attitudes towards Codling Moth Trapping and Suppression Using the Sterile Insect Technique in New Zealand"

_insects, 2019, doi:10.3390/insects10100335_

Round 1

Reviewer 1 Report

The manuscript of Paterson et al describes the results from a questionnaire survey on the public attitude towards codling moth control through trapping and SIT. Overall the study addresses an interesting subject relevant for decision making on pest control methods in urban areas. The text is well written and I suggest publication after minor revision. 

I suggest to address the following points.

Line 18: New technique. I would not consider SIT as a new technique. SIT was developed in the 1930s and has in the 90’s been intensively applied against Cydia pomonella.

Introduction. In my opinion the introduction should inform a bit more about SIT in general. I suggest to give a few details like mentioning mass rearing, sterilization techniques, strength and weakness of the method.

I would also like to see a bit more of background information of C. pomonella in New Zealand/the infected area.

Figure 1. As I understood the Fig is meant to illustrate the area chosen to study the community attitude towards SIT. I don’t find the figure particularly helpful, especially as the insert is small and of limited quality for illustrating the peri-urban aspect. In fact, the larger part of the figure representing the position at the NZ island is pretty irrelevant. Geographical aspects are important for successful application of SIT. Definition of release zones, borders and buffer areas are decisive. Therefore, if providing an aerial overview I would be interested in more specific illustration related to potential SIT application (if such information/detailed planning exists).

Line 191. It appears that the survey participants either have an apple tree (81%) or a walnut tree (19 %) but not both. Is this coincidence or an error?

Line 426. Have replies to question 8 and 9 (harvest and worm in the apple) been represented in the results and discussion? I apologize in case I overlooked that aspect.

Author Response

I note minor revision, responses follow.

The manuscript of Paterson et al describes the results from a questionnaire survey on the public attitude towards codling moth control through trapping and SIT. Overall the study addresses an interesting subject relevant for decision making on pest control methods in urban areas. The text is well written and I suggest publication after minor revision. 

I suggest to address the following points.

Line 18: New technique. I would not consider SIT as a new technique. SIT was developed in the 1930s and has in the 90’s been intensively applied against Cydia pomonella.

New is deleted

Introduction. In my opinion the introduction should inform a bit more about SIT in general. I suggest to give a few details like mentioning mass rearing, sterilization techniques, strength and weakness of the method.

Done

An increase in incursions has led to research on novel technologies to respond to dispersed insects whose exact location is uncertain. Correspondingly, there is an increasing need for understanding of community perceptions in relation to these new technologies [15]. Combinations of benign tactics can offer can offer quicker and more cost effective suppression of pest populations [16,17].

In particular, autocidal methods are preferred where synergy is sometimes possible between methods [18], and in particular the Sterile Insect Technique (SIT) can help to suppress populations in proportion to the mass released and residual wild population [19,20]. A multi-faceted approach including SIT [21] is being evaluated for the local eradication of codling moth in New Zealand (Cydia pomonella) [22]. Orchard populations of this pest have been dramatically reduced by mating disruption over the last decade [12]. A Canadian initiative to develop the SIT for codling moth [23] led to the development of a factory and long-standing release of moths into orchards for population suppression [24]. In New Zealand, we have targeted an isolated sub-region (Central Hawke’s Bay) to test the effects of the SIT on private orchard land. This combination of the SIT using insects imported weekly from British Columbia, together with sex pheromone dispensers for mating disruption covering a series of large (50–100 ha) orchards has greatly reduced populations of codling moth since 2014 [22].

I would also like to see a bit more of background information of C. pomonella in New Zealand/the infected area.

multi-faceted approach including SIT [22] is being evaluated for the local eradication of codling moth in New Zealand (Cydia pomonella) [23], and insect which has been present for 100 years and the target of IPM by mating disruption over large areas (35% market uptake)[12]. Export-focused and documented suppression of codling moth has involved submission of spray diaries and insect counts, and a shift towards more benign control tactics. [12] Orchard populations of this pest have been dramatically reduced by mating disruption over the last decade [12].

Figure 1. As I understood the Fig is meant to illustrate the area chosen to study the community attitude towards SIT. I don’t find the figure particularly helpful, especially as the insert is small and of limited quality for illustrating the peri-urban aspect. In fact, the larger part of the figure representing the position at the NZ island is pretty irrelevant.

Figure revised and simplified.

Geographical aspects are important for successful application of SIT. Definition of release zones, borders and buffer areas are decisive. Therefore, if providing an aerial overview I would be interested in more specific illustration related to potential SIT application (if such information/detailed planning exists).

Fig just shows sampled area now. The comment is ahead of the program, there is no periurban SIT.

In particular, autocidal methods are preferred where synergy is sometimes possible between methods [18], and in particular the Sterile Insect Technique (SIT) can help to suppress populations in proportion to the mass released and residual wild population [19,20]. A multi-faceted approach including SIT [21] is being evaluated for the local eradication of codling moth in New Zealand (Cydia pomonella) [22], and insect which has been present for 100 years and the target of IPM by mating disruption over large areas (35% market uptake)[12]. Orchard populations of this pest have been dramatically reduced by mating disruption over the last decade [12]. A Canadian initiative to develop the SIT for codling moth [23] led to the development of a factory and long-standing release of moths into orchards for population suppression [24]. In New Zealand, we have targeted an isolated sub-region (Central Hawke’s Bay) to test the effects of the SIT on private orchard land. This combination of the SIT using insects imported weekly from British Columbia, together with sex pheromone dispensers for mating disruption covering a series of large (50–100 ha) orchards has greatly reduced populations of codling moth since 2014 [22]. We set out to explore the peri-urban sources.

Line 191. It appears that the survey participants either have an apple tree (81%) or a walnut tree (19 %) but not both. Is this coincidence or an error?

This is correct not a coincidence, correct as written.

Line 426. Have replies to question 8 and 9 (harvest and worm in the apple) been represented in the results and discussion? I apologize in case I overlooked that aspect.

This was brought in with likelihood to care about biosecurity.

I note minor revision

Reviewer 2 Report

In the present paper, Authors carried out a questionnaire survey in the local community of Hastings, New Zealand, to receive the community perception on innovative tools, such as SIT or UAV, used for codling moth control.

The paper has some interesting points and give a view on aspects that are often discarded but can hamper the success of a large scale control program. However, can be improved in some parts, for example a better description in the Introduction about the current codling moth pest status in New Zealand and more information of the local application of SIT.

The main limit I see is that the study seems to have a local interest, in fact it is not clear what is the representativeness of the survey conducted on 86 householder in a single location respect to the more general picture for New Zealand. Also, it should better explained what are the outcomes of this survey, if any, on the ongoing control campaign. For example, was the survey useful to take some decisions or answers from respondents induced to make some adaptations? Also, the treated argument (use of questionnaire survey in the local community to improve area wide pest control programs) should be better framed in an international contest: other examples of similar surveys conducted for other pest control programs are available in the international literature.

Some specific remarks:

Lines 82-84: please explain in more details how you intend to achieve this goal.

Line 231: Predator Free New Zealand 2050: use a reference for this

Line 232: what is “1080”?

Author Response

The comments were useful thanks. Most adopted.

In the present paper, Authors carried out a questionnaire survey in the local community of Hastings, New Zealand, to receive the community perception on innovative tools, such as SIT or UAV, used for codling moth control.

The paper has some interesting points and give a view on aspects that are often discarded but can hamper the success of a large scale control program. However, can be improved in some parts, for example a better description in the Introduction about the current codling moth pest status in New Zealand and more information of the local application of SIT.

Done

The main limit I see is that the study seems to have a local interest, in fact it is not clear what is the representativeness of the survey conducted on 86 householder in a single location respect to the more general picture for New Zealand. Also, it should better explained what are the outcomes of this survey, if any, on the ongoing control campaign. For example, was the survey useful to take some decisions or answers from respondents induced to make some adaptations? Also, the treated argument (use of questionnaire survey in the local community to improve area wide pest control programs) should be better framed in an international contest: other examples of similar surveys conducted for other pest control programs are available in the international literature.

Some specific remarks:

Lines 82-84: please explain in more details how you intend to achieve this goal.

A successful eradication programme requires a holistic and multidisciplinary approach [28], which includes involvement of communities living in close proximity to the pest and potentially within the area targeted for control [29]. In our case, these are peri-urban communities located not far from horticultural enterprises, which we seek to better understand through dialogue.

Line 231: Predator Free New Zealand 2050: use a reference for this

Inserted

Line 232: what is “1080”?

Deleted synonym

Reviewer 3 Report

This is an interesting and well-written paper on community attitudes towards the introduction of SIT and some associated management techniques for the suppression of codling moth.

Overall the paper reads well and is interesting, particularly as it demonstrates the importance of evaluating place-based public responses towards a biocontrol strategy.

General comments:

You need to provide a rationale in the Introduction for measuring 'family history' and the two 'concern for pest species' questions in the survey, as well as expand on the relevance when discussion results - otherwise it's not clear why they were included. The preamble to your questionnaire - and your Info sheet - comprised biased/loaded language: "...which is great because it supports the economy without insecticides". Lines 411-412. While this framng is not ideal, methodologically, it can and should be rationalised in your method, as to why the language was positively framed and not neutral. Question 10 in your questionnaire also has biased language: "... release of harmless sterile moths" and needs similar justification in the methodology. Line 428. Data for Questions 13 and 16 (open-ended responses) are valuable social data and should be coded & presented in the Results section 3.6, perhaps as a table with themes and representative quotes. I notice you have included a table of questions raised (Table 1), but I would like to see the qualitative data in the main body of the paper, because you have obviously used it to make your assessments throughout the Discussion (e.g. Line 247, "... it was evident from open-ended questions", and especially in Section 3.6 of the Results. Without this data included in the paper, you cannot refer to it and all you can refer to are the very basic descriptive questions from the survey. Lines 273-277: needs unpacking a bit more re possible solutions or suggested recommendations for responding to historical problems. Perhaps tone down the language used in Lines 232-233 - the community as a whole did not support, only a majority of the selected response sample. In the Introduction (Lines 87-93) and in the Discussion/Conclusion, there needs to be some acknowledgement of the viability of this type of SIT approach in the broader market context - this relates to its utility. Unless the fruit being protected is only for domestic use in NZ, there needs to be a mention or caution about market access considerations for the acceptability of fruit using SIT, as not all markets will accept SIT as a control measure. This will demonstrate that the questions in this study is relevant to the broader horticultural context - particularly as exporters are funding the application of SIT in the region. Finally, just a comment, it was really interesting to read the findings presented in Section 3.4 and perhaps future research could explore/test the implications of these control methods (i.e. incentives, exchanges, behavioural tolerances). 

Author Response

Thanks for the comments, mostly adopted.

This is an interesting and well-written paper on community attitudes towards the introduction of SIT and some associated management techniques for the suppression of codling moth.

Overall the paper reads well and is interesting, particularly as it demonstrates the importance of evaluating place-based public responses towards a biocontrol strategy.

General comments:

You need to provide a rationale in the Introduction for measuring 'family history' and the two 'concern for pest species' questions in the survey, as well as expand on the relevance when discussion results - otherwise it's not clear why they were included.

Done

Questions included characterisation of gender, age and because of an interest in the spread of information in the horticulture sector about biosecurity, whether the respondent had family that worked in horticulture. We pursued this with the respondent’s level of concern about mammalian pests and invasive insect species, whether they were happy to have monitoring traps set up, whether there were any apple or walnut trees on the property, and their acceptance of the use of unmanned aerial vehicles for insect release.

The preamble to your questionnaire - and your Info sheet - comprised biased/loaded language: "...which is great because it supports the economy without insecticides".

Lines 411-412. While this framng is not ideal, methodologically, it can and should be rationalised in your method, as to why the language was positively framed and not neutral.

See line 147 The fact that these harmless practices support the economy and avoid the use of insecticides was also highlighted

Question 10 in your questionnaire also has biased language: "... release of harmless sterile moths" and needs similar justification in the methodology.

Done

Line 428. Data for Questions 13 and 16 (open-ended responses) are valuable social data and should be coded & presented in the Results section 3.6, perhaps as a table with themes and representative quotes. I notice you have included a table of questions raised (Table 1), but I would like to see the qualitative data in the main body of the paper, because you have obviously used it to make your assessments throughout the Discussion (e.g. Line 247, "... it was evident from open-ended questions", and especially in Section 3.6 of the Results. Without this data included in the paper, you cannot refer to it and all you can refer to are the very basic descriptive questions from the survey.

This is done Table 1, and the 11 respondents are reported by gender and all had early concerns addressed. The Table presents the material sought.

3.1. Age and gender

According to the most recent census (2013), 52% of Hastings residents are female and 48% male [32]. Of the 86 respondents, 63% (n= 54) were female and 39% (n=32) were male. The greater response from females was a reflection of presence in the households surveyed and willingness to participate.  One male and ten females expressed initial reservations but all were satisfied with the final response to open questions.

Lines 273-277: needs unpacking a bit more re possible solutions or suggested recommendations for responding to historical problems.

Perhaps tone down the language used in Lines 232-233 - the community as a whole did not support, only a majority of the selected response sample.

Done

In the Introduction (Lines 87-93) and in the Discussion/Conclusion, there needs to be some acknowledgement of the viability of this type of SIT approach in the broader market context - this relates to its utility. Unless the fruit being protected is only for domestic use in NZ, there needs to be a mention or caution about market access considerations for the acceptability of fruit using SIT, as not all markets will accept SIT as a control measure.

This is factually incorrect so not changed. Fruit is accepted in all markets tested.

This will demonstrate that the questions in this study is relevant to the broader horticultural context - particularly as exporters are funding the application of SIT in the region.

This should make it obvious that is works and is cost-effective ?

Finally, just a comment, it was really interesting to read the findings presented in Section 3.4 and perhaps future research could explore/test the implications of these control methods (i.e. incentives, exchanges, behavioural tolerances). 

rewritten

3.6. General community response to the survey

Open-ended comments were recorded at the end of the survey, including from ten women and one man expressing early reservations. From the responses it is clear that most people surveyed were very supportive of the proposed eradication methods once they understood the technical details sufficiently to judge them benign. Many people voiced concerns over health issues such asthma and eczema as a result of insecticide spraying in close proximity to their homes and were, therefore, very supportive of sterile insect releases. Some members of the community preferred the sterile insect method for ethical reasons as no organisms are directly killed. Questions and concerns were raised about whether, or how, the proposed actions would affect respondents’ pets or children. It was explained that the proposed methods should not affect them at all. This explanation apparently eased reservations about the proposed methods

Reviewer 4 Report

A well written paper on an increasingly important facet of pest management. Congratulations to the authors for seeking information upon which to guide future efforts in pest management in urban and peri-urban areas. However, I think the title is slightly misleading and would suggest the inclusion of ‘releases by unmanned aerial vehicle’ given the quantification of study participant responses in this area, but not in response to SIT. I would also suggest the authors read and incorporate findings from the following papers:

Mankad, A. (2016) Psychological influences on biosecurity control and farmer decision-making. A review. Agronomy for Sustainable Development 36: 40. https://doi.org/10.1007/s13593-016-0375-9

Tapsuwan, S, Tam, M, Capon, T, Whitten, S, Kandulu, J and Measham PF. (2019) Willingness to pay for area-wide management and sterile insect technique to control fruit flies in Australia, International Journal of Pest Management DOI: 10.1080/09670874.2019.1652369

Some specific suggestions;

Line 51 – exact location being uncertain – please expand and introduce the concept of area-wide management here (AWM). There is no definition of AWM throughout the paper, which I feel is a key area that would add context to this work.

Line 63 – community contribution to AWM – suggest you look to Tapsuwan et al 2019

Line 84 – please clarify your definition of peri-urban.

Line 86 – include reference for withdrawal of methyl bromide

Line 92 – insert ‘management’ between area-wide and programme

Line 129-130 – further clarification; selection of dwellings within set parameters appears to be in conflict with ‘random’ selection comment in Line 104

Line 194-195 – different representation of numbers of participants – please be consistent

Line 198 (and section 3 in general) - explanation provided to study participants that SIT moths would be released by UAVs, but not an explanation of sterile insect technique? Comments recorded by participants regarding SIT but need to justify why this aspect was not quantified?

Line 273-277 – the last paragraph in Section 4 does not add value; the last comment is not referenced nor was this a part of the current study.  I suggest deleting.

Author Response

Thanks for the comments, the majority have been adotped.

A well written paper on an increasingly important facet of pest management. Congratulations to the authors for seeking information upon which to guide future efforts in pest management in urban and peri-urban areas. However, I think the title is slightly misleading and would suggest the inclusion of ‘releases by unmanned aerial vehicle’ given the quantification of study participant responses in this area, but not in response to SIT.

The potential for ground release is unrealistic due to cost. I have not adopted this recommendation.

I would also suggest the authors read and incorporate findings from the following papers:

Mankad, A. (2016) Psychological influences on biosecurity control and farmer decision-making. A review. Agronomy for Sustainable Development 36: 40. https://doi.org/10.1007/s13593-016-0375-9

Tapsuwan, S, Tam, M, Capon, T, Whitten, S, Kandulu, J and Measham PF. (2019) Willingness to pay for area-wide management and sterile insect technique to control fruit flies in Australia, International Journal of Pest Management DOI: 10.1080/09670874.2019.1652369

These papers have now been cited.

Some specific suggestions;

Line 51 – exact location being uncertain – please expand and introduce the concept of area-wide management here (AWM). There is no definition of AWM throughout the paper, which I feel is a key area that would add context to this work.

Line 63 – community contribution to AWM – suggest you look to Tapsuwan et al 2019

Line 84 – please clarify your definition of peri-urban.

A successful eradication programme requires a holistic and multidisciplinary approach [28], which includes involvement of communities living in close proximity to the pest and potentially within the area targeted for control [29]. In our case, these are peri-urban communities located not far from horticultural enterprises, which we seek to better understand.

Line 86 – include reference for withdrawal of methyl bromide

If successful, controls of codling moth in apple orchards targeting export markets would not be needed, nor would fumigation for market access (as an ozone depleting gas, fumigant methyl bromide is to be withdrawn in 2021, under the 1987 Montreal Protocol).

Line 92 – insert ‘management’ between area-wide and programme

Done

Line 129-130 – further clarification; selection of dwellings within set parameters appears to be in conflict with ‘random’ selection comment in Line 104

The parameters selected were for the study area.

Line 194-195 – different representation of numbers of participants – please be consistent

They are a subset, explained.

Line 198 (and section 3 in general) - explanation provided to study participants that SIT moths would be released by UAVs, but not an explanation of sterile insect technique?

This is incorrect – see Appendix for explanation used at the start on SIT

Comments recorded by participants regarding SIT but need to justify why this aspect was not quantified?

In particular, autocidal methods are preferred where synergy is sometimes possible between methods [18], and in particular the Sterile Insect Technique (SIT) can help to suppress populations in proportion to the mass released and residual wild population, through the mass rearing and release of sterile insects [19-21].

Line 273-277 – the last paragraph in Section 4 does not add value; the last comment is not referenced nor was this a part of the current study.  I suggest deleting.

It was referenced and part of the study - Table 1.

Round 2

Reviewer 3 Report

The revisions have significantly improved the paper, thank you for taking on feedback.

Author Response

(x) English language and style are fine/minor spell check required

The revisions have significantly improved the paper, thank you for taking on feedback.

there were no comments to address but the text is yellow?